# The neural influence of autobiographical memory related to the parent-child relationship on psychological health in adulthood

Eun Seong Kim[1,2☯], Hesun Erin Kim[1,3☯], Jae-Jin Kim[1,3,4]*

1 Institute of Behavioral Science in Medicine, Yonsei University College of Medicine, Seoul, Republic of Korea, 2 Department of Occupational Therapy, Chunnam Techno University, Gokseong, Republic of Korea, 3 Brain Korea 21 PLUS Project for Medical Science, Yonsei University College of Medicine, Seoul, Republic of Korea, 4 Department of Psychiatry, Yonsei University Gangnam Severance Hospital, Seoul, Republic of Korea

☯ These authors contributed equally to this work.
* jaejkim@yonsei.ac.kr

**Data Availability Statement:** All relevant data are within the manuscript and its Supporting Information files.

## Abstract

The recollection of childhood memories is affected by the subjects involved, such as father and mother, and by the context. This study aimed to clarify the neural influence of autobiographical memory related to the parent-child relationship on psychological health in adulthood. Twenty-nine healthy volunteers participated in a functional magnetic resonance imaging experiment using a childhood memory recollection task, in which they appraised the emotion a parent would have provided in a given situation. Whole-brain univariate and psychophysiological interaction analyses were performed. Neuroimaging results indicated notable involvement of the caudal anterior cingulate cortex and precuneus in autobiographical memory related to the parent-child relationship, and their activities were closely associated with the level of depression and self-esteem, respectively. The functional connectivity results indicated increased connectivity between the caudal anterior cingulate cortex and fusiform gyrus for the father-positive condition compared to the mother-positive condition and there was a positive correlation between the strength of connectivity between the two regions and the anxiety level. Our findings suggest the processing of negative affect and the personalness of autobiographical memories are distinctly engaged depending on the parent in question and the situational valence. The present study illuminates the impact of autobiographical memory processes on various dimensions of psychological health.

## Introduction

Like the saying, "no health without mental health," promoted by the WHO, psychological health or psychological well-being is a fundamental factor contributing to quality of life [1]. There are many factors that influence psychological health in adulthood, and one of the most central determinants is the parent-child relationship [2]. The relationship between a

**Funding:** This work was supported by the National Research Foundation of Korea (NRF) grant funded by the Korea government (MSIP) (No. NRF-2016R1A2A2A10921744).

**Competing interests:** The authors have declared that no competing interests exist.

parent and child becomes the template for other interpersonal relationships because attachment bonding allows a person to understand their environment and establish a sense of security [3,4]. An examination of the association between the retrospective perceptions of the parent-child relationship and emotional health in adulthood showed that daily emotional and psychological distress are closely related to how one perceives the quality of the parent-child relationship, and the retrospective perceptions of father-child and mother-child relationships contribute to different dimensions of emotional health [5]. Therefore, the perceptions of the childhood relationship with parents are important in the overall well-being, even as adults.

The retrospective perceptions of the parent-child relationship are based on an individual's autobiographical memory. The development of autobiographical memory is a social process that is influenced by the reminiscing style of parents, and thus an integral part of the developing sense of self and interpersonal relatedness [6,7]. Autobiographical memory represents the self and shapes emotional states of our everyday lives [8,9]. In general, memories are highly intertwined with features like self-referential processing and emotion [9]. Functional neuroimaging studies have examined autobiographical memories in reference to other types of memories. For instance, a recent study that compared the difference between autobiographical memory and simple recognition memory found that when a task required retrieval of life memories, default-mode areas were engaged more, whereas a simple recollection of memories engaged the parietal memory network [10]. Since autobiographical memory retrieval relies heavily on self-reference and emotion, retrieval of autobiographical memories has been shown to employ various cortical and subcortical regions, including the prefrontal and parietal cortices, anterior cingulate cortex (ACC), precuneus, medial temporal cortex, hippocampus, and amygdala [11–14]. In particular, the amygdala has been implicated in the encoding of emotional memories [15] as well as retrieval of autobiographical memories [16]. Previous studies also found coactivation of the amygdala, hippocampus, and inferior frontal gyrus (IFG) during the retrieval of autobiographical memories [17] and neural responses in the dorsal to caudal ACC and IFG elicited by the retrieval of painful autobiographical memories [18]. These reports consistently show a close relationship between autobiographical memory and emotional memory.

Psychological health is very closely related to memory encoding and retrieval, and there is evidence supporting its influence on neural activities related to autobiographical memory. Numerous studies have also shown the connection between psychological health and selective memory bias. The representation, recall, and maintenance of autobiographical memories have shown to be disturbed by individuals with poor psychological well-being, where they are systematically biased for negative information [11]. Memory bias and negative interpretations of events contribute to a vicious cycle, where one's psychological health contributes to the tendency toward negative interpretation of memories, and this bias also maintains low self-esteem, depression, or anxiety [19–21]. In fact, there is a consensus that self-esteem, anxiety, and depression are core elements of psychological health [22–24]. The importance of these elements has spurred researchers to study the close associations between self-esteem, anxiety and depression, and quality of life in numerous fields [25–28].

Self-esteem is a facet of personality that affects perception of social standing and modulates the salience of social interaction. Previous studies have shown that the midline cortical structures, such as the medial prefrontal cortex (PFC), ACC, posterior cingulate cortex (PCC), and precuneus that mediate self-referential processing are also closely involved in the expression of self-esteem [29]. Global self-esteem, which includes self-competence and self-liking, is related to selective memory, especially toward a negatively salient stimulus [19]. Individuals with lower self-esteem tend to be more concerned with how others perceive them, and this makes

heightened negative events more memorable. Lower self-esteem predicts an increase in ventral ACC and medial PFC activity in response to positive versus negative social feedback, indicating the relationship between self-esteem and the salience processing [30]. On the other hand, a positivity bias for events that involve self-evaluations is stronger for individuals with higher self-esteem, suggesting that this bias affecting autobiographical memory is part of psychological mechanisms for maintaining a positive self-image [31]. The underlying neural substrates of a positive self-evaluation include the medial ventral and dorsolateral PFC and hippocampus, reflecting cognitive effort and emotional involvement for the positivity bias [32]. In addition, it has been reported that the lateral PFC, dorsal ACC, posterior cingulate cortex (PCC), precuneus, and caudate are involved in negative or positive character feedback associated with self-esteem [33].

The level of self-esteem may be related to the manifestation of depression through memory biases [34]. Many studies have reported that overall autobiographical memory predicts the onset or course of depression and thus is considered to be a risk factor for depression [35,36]. Depression contributes to the deviant reactivation of neural areas associated with autobiographical memory retrieval [37]. Neural correlates of autobiographical memory deficits in depressed patients have been known to involve a wide range of brain regions, such as the IFG, lateral orbitofrontal cortex (OFC), dorsomedial and ventrolateral PFC, ACC, precuneus, temporoparietal cortex, insula, hippocampus, and amygdala [38–40].

Pathological anxiety has also been associated with negative memory bias [41]. Like depression, anxiety disorders are also characterized by maladaptive emotional responses and retrieval of memories [42]. In particular, negative autobiographical memory bias is prominent in individuals high in social anxiety [43]. Increased state anxiety decreases the ability to retrieve specific autobiographical memory [44]. Previous studies of patients with anxiety disorders have reported functional impairments of structures linked to the experience and regulation of emotion, such as the PFC, ACC, insula, and amygdala [45–47].

Although behavioral and neurobiological studies indicate the impact of the psychological dimensions on the encoding and retrieval of autobiographical memories, the way in which psychological health affects memory processes in the general population is still unknown. The importance of the parent-child relationship in child development has been well documented. When looking at the way parents interact with a child, mothers are often more caring and nurturing, whereas fathers engage in more playful interactions [48]. Naturally, since the relationship between a father and child is different from the relationship between a mother and child, the autobiographical memories of parents in adulthood must be different. This difference is also observed in the brain response. For example, faces of mothers elicited more activity in the face processing network including the fusiform gyrus, whereas faces of fathers elicited more activity in the striatal region [49]. Despite the importance of the parent-child relationship and its difference between parents, no study has provided neurobiological evidence for how perceptions change depending on the parent and the valence of memories.

In this study of the general population, we sought to identify the neural substrates of the impression of parents while participants recalled childhood memories. For this purpose, we examined how fathers and mothers are perceived differently and whether the valence of memory influences this perception, and surveyed the effects of psychological health (self-esteem, depression, and anxiety) on the memory process. As childhood memories of parents triggered by various situations are inherently charged with emotions, we hypothesized that regions often involved in an autobiographical memory, such as the ACC, IFG, amygdala, and hippocampus, would be engaged differently depending on the parent and the valence of situation, and would be modulated by a person's psychological health.

## Methods

### Participants

Twenty-nine normal volunteers were recruited through an advertisement at a local hospital and via the internet. All participants were right-handed, as assessed by the Annett Handedness Inventory [50]. Exclusion criteria included the presence of a neurological, psychiatric or significant medical illness, and a history of current or past substance abuse or dependence. Two participants who did not complete the behavior task were excluded, and three were excluded due to framewise displacement greater than 3 mm. A total of 24 participants (age: 28.0 ± 7.4 years old; 11 males and 13 females; education: 15.8 ± 4.0 years) were included in the final analysis. The study was approved by the Institutional Review Board of Yonsei University Severance Hospital, and written informed consent was obtained from all participants before the study began.

### Psychological assessment

We used the Parent-Adolescent Communication Inventory (PACI) [51], which consists of ten items assessing the openness of family communication and ten items measuring the negative aspects of interaction and communication among family members on a five-point scale (1 = not like that at all, 5 = always the case), to measure participants' quality of communication with their parents. The ten-item Rosenberg Self-Esteem Scale (RSES) [52] was also used to measure participants' self-respect on a four-point scale (1 = strongly disagree, 4 = strongly agree). We used the Hospital Anxiety and Depression Scale (HADS) [53], which is comprised of seven items for anxiety and seven items for depression with each scored from 0 to 3 (0 to 21 for either anxiety or depression total scores), to determine the levels of anxiety and depression.

### Task stimuli

Participants performed a task where they were asked to imagine what their parents' facial expression looked like in a given situation during their childhood. Visual stimuli with a grey background included black words presenting a situation in the upper part and four black and white faces of the same person expressing different emotions (very happy, a little happy, very angry, and a little angry) in the lower part (Fig 1). The number of situations used in the task was 30, including ten positive (e.g., when I did errands), ten negative (e.g., when I did not do homework), and ten neutral (e.g., when I was on the phone). All actual situations used are shown in S1 Table.

In order to evaluate the validity of the presented words, the vividness of memory associated with the situation, the intensity of reaction received from a parent in the situation, and the current degree of arousal caused by the recollection of the situation were assessed for each situation in 24 volunteers who did not participate in the fMRI experiment (27.3 ± 1.7 years old; 13 males and 11 females). Similar to the fMRI task, each situation was presented with either word "Father" or "Mother" on top of a page, then the volunteers were instructed to visualize an image of a parent during their childhood. All three items (vividness, response intensity, and arousal) were measured on a 9-point scale (0: not at all, 4: moderately, and 8: extremely). Analysis was conducted using repeated measures analysis of variance (ANOVA) for parent (Father and Mother) and valence of situation (Positive, Negative, and Neutral) and post-hoc paired $t$-test.

### Behavioral task and analysis

A total of ten blocks were presented in the task, and each block lasted for 30 sec with a 16 sec rest period between the blocks. Half of the blocks included a male face and the other half of the blocks included a female face. Participants were instructed to regard the man as their father

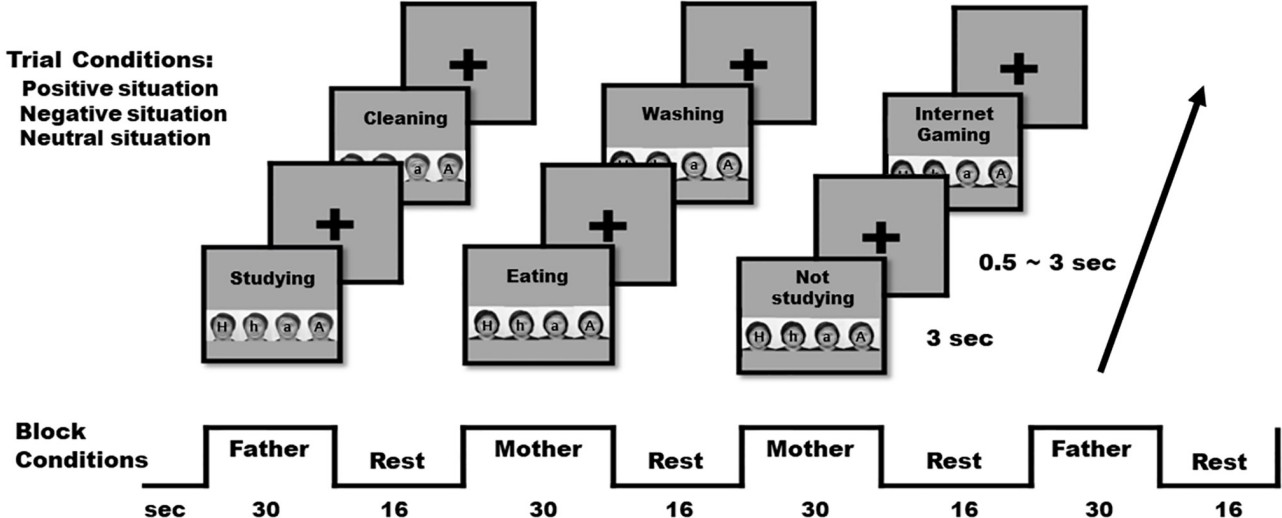

**Fig 1. Task stimuli and design.** The task imagining what one's parents' facial expression looked like in a certain situation of his/her childhood was built with mixed design, in which there were two experimental blocks for the father and mother conditions and six trials (two positive, two negative, and two neutral situations) in each block. The facial expressions presented were of four types: very happy (H), a little happy (h), a little angry (a), and very angry (A).

and the woman as their mother. The 30 situations were presented in five blocks for the father condition and all of these situations were repeated in five blocks for the mother condition. Each block included two positive, two negative, and two neutral trials. Each trial lasted for 3 sec and the inter-stimulus interval was jittered between 0.5 sec and 3.0 sec. We randomly assigned the two block conditions in the whole sequence, the six situations in each block, and the four facial emotions in each trial. The timing and distribution of trials were optimized by optseq2 (http://surfer.nmr.mgh.harvard.edu/optseq/) to ensure that events in this mixed design could be statistically separated and to verify that the neural responses for each pair of stimuli were mutually orthogonal.

Participants responded by pressing one of four buttons corresponding to the location of the face that contained the closest imagined expression of their father or mother in the presented situation. The emotions of the selected faces were converted to scores (2 = very happy, 1 = a little happy, -1 = a little angry, and -2 = very angry) to represent the imagined emotion. In order to verify the validity of facial stimuli, we asked participants who finished scanning to rate the valence of each face on a 4-point Likert scale, anchored at 2 = very positive and -2 = very negative.

The imagined emotion score and reaction time (RT) were counted for behavioral analysis. The ANOVA was used to identify the main effect of parents (father and mother) and situation (positive, negative, and neutral) and the interaction effect between them. A paired *t*-test was subsequently performed to investigate the difference in imagined emotion scores between the father and mother conditions in each of the positive, negative, and neutral situations.

## Imaging parameters and processing

MRI data were acquired on a 3T scanner (Achieva; Philips Medical System, Best, Netherlands). Functional images were acquired using a T2*-weighted gradient echo-planar imaging sequence (31 axial slices × 3 mm thickness and no gap; repetition time [TR] = 2,000 ms; echo time [TE] = 30 ms; flip angle [FA] = 90˚; in-plane matrix = 124 × 124 pixels; and field of view

[FOV] = 220 mm). Imaging slices were obtained at a tilted angle of 30˚ from the anterior commissure-posterior commissure line to minimize signal loss in the orbitofrontal cortex. An additional structural image using a 3D T1-weighted gradient echo sequence (TR = 9.692 ms; TE = 4.59 ms; FA = 8˚; image matrix = 224 × 224; and FOV = 220 mm) was obtained from each participant after the functional runs.

Data preprocessing and analysis were performed using SPM12 (Wellcome Department of Cognitive Neurology, University College London). Images remaining after discarding the first six images from the dummy scan were used for further preprocessing. Differences in the slice acquisition time were corrected, and realignment was performed to correct for head motion. These functional images were coregistered on the T1-weighted image. The T1-weighted images were spatially normalized to the standard template, and the resulting transformation matrices were applied to the coregistered functional images. The normalized images were smoothed with a Gaussian kernel of 6 mm full-width at half-maximum.

### Imaging data analysis

Preprocessed images were analyzed using a general linear model. A series of events as regressors of interest was modeled by convolving the event-train of stimulus onsets with the canonical hemodynamic response function. Trials were modeled based on the onset of stimulus presentation, specified as zero-duration events. With movement parameters as regressors of no-interest, left-hand and right-hand button pressing during the task period were also included as additional regressors. The positive and negative situations were contrasted with the neutral situations in each of the father and mother conditions for each participant on first-level analysis. At this first stage of analysis, the neutral situation was chosen as a control to subtract out general effects involving visual process and recall. Subsequently, four main contrasts for the positive and negative emotional situations in the father and mother conditions were created and used for the second-level analysis using a flexible factorial model and a *post-hoc* paired *t*-test to investigate the main and interaction effects of parents and situation. Statistical inferences were conducted at a threshold of a corrected $p < 0.05$, which corresponds to the family-wise error (FWE) corrected significance at the cluster level with a cluster-defining threshold of $p < 0.001$. In addition, correlation analysis was conducted between parameter estimates from the significant clusters showing the interaction effect and psychological scale scores. Parameter estimates were extracted from 6-mm spherical regions of interest (ROIs) centered on the peak voxel of significant clusters using the MarsBaR toolbox.

Functional connectivity analyses were implemented with psychophysiological interaction (PPI) analyses, in which the seed region was defined as a cluster showing the interaction effect in the second-level analysis. The time course from a 6-mm sphere in each seed region was extracted. Results were reported at a cluster-level FWE-corrected $p < 0.05$, with an initially voxel-level uncorrected $p < 0.001$. Finally, correlation analyses were conducted between the strength of functional coupling of the seed with the coupled region and the psychological scale scores. The MarsBaR toolbox was used to extract mean PPI coefficients from the first-level single-subject contrasts using the target area used for PPI analyses.

## Results

### Stimulus assessment results

Results from the assessment of the vividness, response intensity, and arousal of the 30 word stimuli are presented in Table 1. The vividness showed a significant result in a main effect of parent ($F_{1,23} = 7.63$, $p = 0.011$), but not in a main effect of situation ($F_{2,46} = 0.46$, $p = 0.637$) and an interaction effect ($F_{2,46} = 1.08$, $p = 0.347$). In post-hoc analysis, the vividness was significantly

**Table 1. Results from the assessment of the vividness, response intensity, and arousal of the 30 word stimuli by the type of the parent and valence (mean ± standard deviation).**

|  | Positive | Negative | Neutral |
|---|---|---|---|
| Vividness |  |  |  |
| Father | 4.92 ± 2.03 | 4.95 ± 1.92 | 5.00 ± 1.75 |
| Mother | 5.65 ± 1.66 | 5.83 ± 1.50 | 5.54 ± 1.60 |
| Response intensity |  |  |  |
| Father | 4.57 ± 1.37 | 4.62 ± 1.11 | 4.23 ± 1.13 |
| Mother | 5.48 ± 1.08 | 5.47 ± 1.24 | 4.78 ± 1.25 |
| Arousal |  |  |  |
| Father | 3.92 ± 1.70 | 3.97 ± 1.70 | 3.64 ± 1.43 |
| Mother | 4.71 ± 1.72 | 4.70 ± 1.63 | 4.15 ± 1.74 |

greater in the mother condition than in the father condition ($t = 2.76$, $p = 0.011$). The response intensity showed significant results in a main effect of parent ($F_{1,23} = 24.23$, $p < 0.001$) and a main effect of situation ($F_{2,46} = 6.83$, $p = 0.003$), but not in an interaction effect ($F_{2,46} = 1.45$, $p = 0.246$). Post-hoc analysis revealed that the response intensity was significantly higher in the mother condition than in the father condition ($t = 4.92$, $p < 0.001$) and significantly higher in the positive situation and negative condition than in the neutral situation ($t = 3.24$, $p = 0.004$; $t = 3.95$, $p = 0.001$, respectively). The arousal showed significant results in a main effect of parent ($F_{1,23} = 6.66$, $p = 0.017$) and a main effect of situation ($F_{2,46} = 5.62$, $p = 0.007$), but not in an interaction effect ($F_{2,46} = 0.98$, $p = 0.384$). In post-hoc analysis, the arousal was significantly higher in the mother condition than in the father condition ($t = 2.58$, $p = 0.017$), and in the positive and negative conditions than in the neutral condition ($t = 3.35$, $p = 0.003$; $t = 3.25$, $p = 0.004$, respectively).

## Behavioral results

Fig 2 illustrates the mean imagined emotion score and mean RT in each condition during the task. The imagined emotion scores showed a main effect of situation [$F_{2,47} = 408.28$, $p < 0.001$], but not a main effect of parents [$F_{1,71} = 0.45$, $p = 0.504$] and a significant

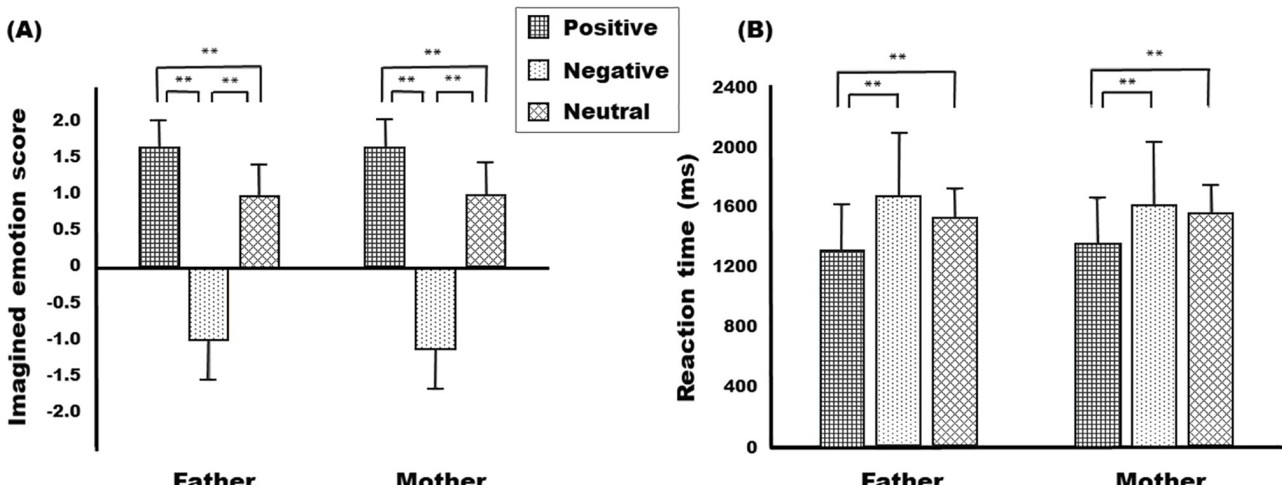

**Fig 2. The imagined emotion scores (A) and reaction times (B) in each situation and each block condition.** $^{**}p < 0.01$.

parents × situation interaction [$F_{2,23} = 0.80$, $p = 0.460$]. Post-hoc analyses showed that the scores for the positive situations were significantly higher than those for the negative ($t = 20.49$, $p < 0.01$) and neutral situations ($t = 14.53$, $p < 0.01$), and the scores for the negative situations were significantly lower than those for the neutral situations ($t = -15.86$, $p < 0.01$).

The RTs also showed a main effect of situation [$F_{2,47} = 45.42$, $p < 0.001$] and a significant parents × situation interaction [$F_{2,23} = 4.42$, $p = 0.024$], but not a main effect of parents [$F_{1,71} = 0.06$, $p = 0.813$]. Post-hoc analyses showed that the RTs for the positive situations were significantly shorter than those for the negative situations ($t = -6.82$, $p < 0.01$) and the neutral situations ($t = -6.77$, $p < 0.01$), and there was no significant difference between the negative and neutral situations ($t = 1.49$, $p = 0.151$). The RTs in the father-negative condition were significantly longer than those in the father-neutral condition ($t = 3.90$, $p < 0.01$), whereas there was no significant difference between the mother-negative and mother-neutral conditions ($t = 1.49$, $p = 0.453$).

In the psychological assessments, the mean openness and problem scores for father in the PACI were 31.3 ± 7.9 and 25.4 ± 5.6, respectively, and those for mother were 38.3 ± 8.9 and 23.3 ± 5.6, respectively. The mean RSES score was 24.9 ± 2.4. The mean anxiety and depression scores in the HADS were 5.2 ± 3.3 and 5.1 ± 3.6, respectively. With regard to the relationships between the imagined emotion scores and PACI, the imagined emotion scores in the father-positive condition were positively correlated with the openness scores for father ($r = 0.45$, $p = 0.028$) and negatively correlated with the problem scores for father ($r = -0.50$, $p = 0.013$), and the imagined emotion scores in the mother-positive condition were also positively correlated with the openness scores for mother ($r = 0.66$, $p < 0.001$) and negatively correlated with the problem score for mother ($r = -0.58$, $p < 0.01$). With regard to relationships with the other scale scores, the only significant result was a positive correlation between the imagined emotion scores in the father-positive condition and the RSES scores ($r = 0.44$, $p = 0.034$).

## Imaging results

As shown in Table 2, four main contrasts for the positive and negative emotional situations in the father and mother conditions showed significant main and interaction effects. The main effect of parents was evident in the left inferior frontal gyrus and left calcarine cortex, in which regional activity was higher in the mother condition than in the father condition. The main effect of situation was found in the left inferior frontal gyrus, left dorsal ACC, bilateral premotor cortex, and left thalamus, in which regional activity was higher in the positive situation than in the negative situation, and the right fusiform gyrus and left insula, in which regional activity was higher in the negative situation than in the positive situation.

There was an interaction effect in the right caudal ACC and right precuneus. As shown in Fig 3A, right caudal ACC activity in the father condition was significantly greater in the negative situation than in the positive situation ($t = 5.09$, $p < 0.01$), whereas that in the mother condition was not significant but greater in the positive situation than in the negative situation ($t = 1.34$, $p = 0.193$). Fig 3B also shows that right precuneus activity in the father condition was significantly greater in the negative situation than in the positive situation ($t = 2.35$, $p < 0.05$), whereas that in the mother condition was greater at a marginal level in the positive situation than in the negative situation ($t = 1.94$, $p = 0.065$). Correlation analysis between regional activity and psychological scale scores revealed a significant negative correlation between right caudal ACC activity in the mother-positive condition and depression scores ($r = -0.45$, $p = 0.029$) and between right precuneus activity in the father-negative condition and RSES scores ($r = -0.44$, $p = 0.033$).

**Table 2. Brain regions showing the main and interaction effects among four main contrasts for the positive and negative situations in the father and mother conditions.**

| Region | BA | MNI coordinates | | | Number of voxels | F | Post-hoc |
|---|---|---|---|---|---|---|---|
| | | x | y | z | | | |
| Main effect of parents | | | | | | | |
| L. inferior frontal gyrus | 44 | -50 | 14 | 20 | 215 | 24.76 | Fa < Mo |
| L. calcarine cortex | 17 | -10 | -90 | 0 | 749 | 31.76 | Fa < Mo |
| Main effect of situation | | | | | | | |
| L. inferior frontal gyrus | 47 | -34 | 22 | -24 | 121 | 22.02 | Po > Ne |
| L. dorsal ACC | 32 | -2 | 24 | 36 | 312 | 22.63 | Po > Ne |
| L. premotor cortex | 6 | -38 | -4 | 58 | 1519 | 29.57 | Po > Ne |
| R. premotor cortex | 6 | 48 | 8 | 34 | 697 | 26.48 | Po > Ne |
| R. fusiform gyrus | 18 | 38 | -74 | -14 | 18460 | 147.75 | Po < Ne |
| L. insula | | -32 | 20 | 6 | 132 | 24.02 | Po < Ne |
| L. thalamus | | -2 | -20 | 12 | 132 | 19.29 | Po > Ne |
| Interaction: parents x situation | | | | | | | |
| R. caudal ACC | 32 | 6 | 10 | 40 | 562 | 30.47 | Fig 3A |
| R. precuneus | 7 | 12 | -50 | 46 | 296 | 17.45 | Fig 3B |

BA, Brodmann area; Montreal Neurological Institute (MNI); L., left; R., right; Fa, father condition; Mo, mother condition; Po, positive emotional situation; Ne, negative emotional situation; and ACC, anterior cingulate cortex.

Significant clusters were obtained at voxel level $P_{unc.} < .001$ and cluster level $P_{FWE} < .05$.

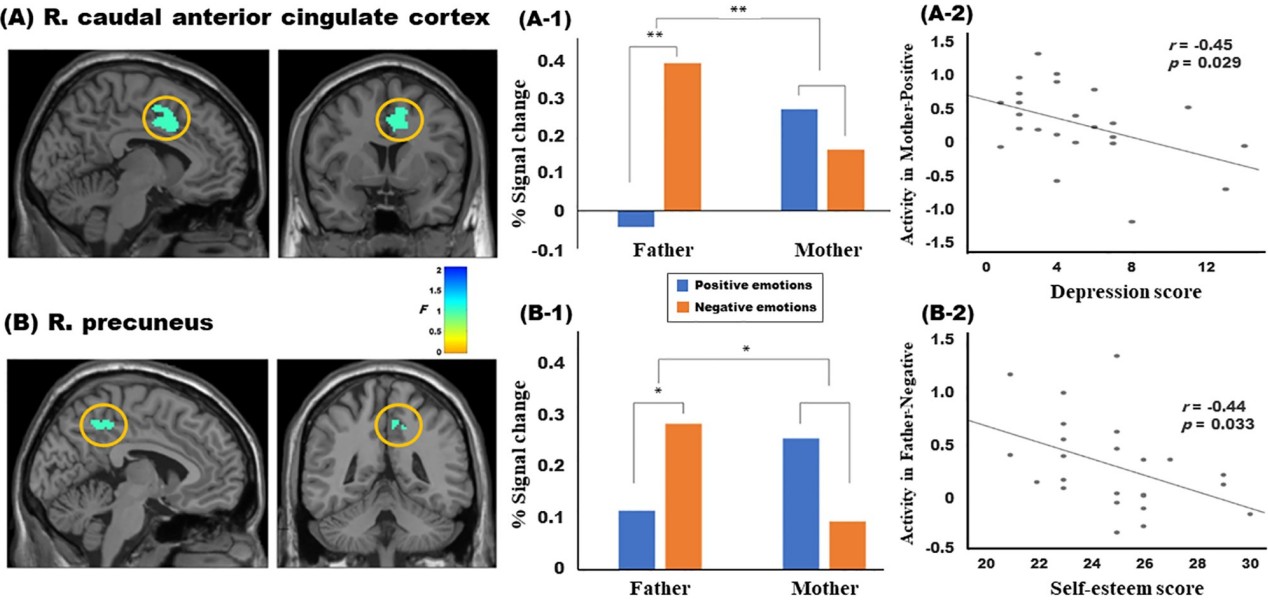

**Fig 3. Brain activity in the right (R.) caudal anterior cingulate cortex (A) and precuneus (B) showing the interaction effect between parents and emotions.** Post-hoc analysis showed that activity in both regions was significantly greater in the father condition than in the mother condition, and that in the father condition was significantly greater in the negative emotions than in the positive emotions (A-1 and B-1). In addition, R. caudal anterior cingulate cortex activity in the mother-positive condition was negatively correlated with the depression scores (A-2), whereas R. precuneus activity in the father-negative condition was negatively correlated with the Rosenberg self-esteem scale scores (B-2). * $p < 0.05$, ** $p < 0.01$.

**Bilateral fusiform gyrus**

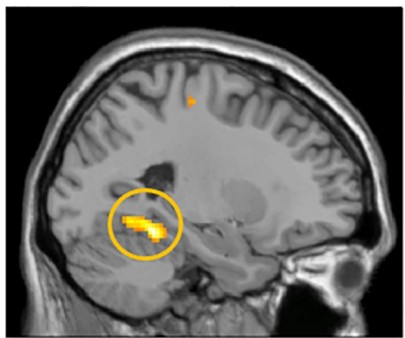
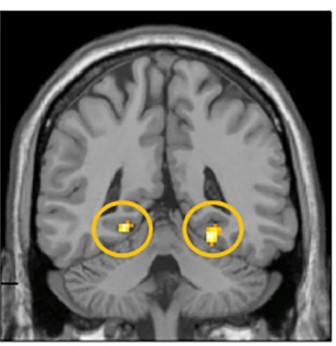
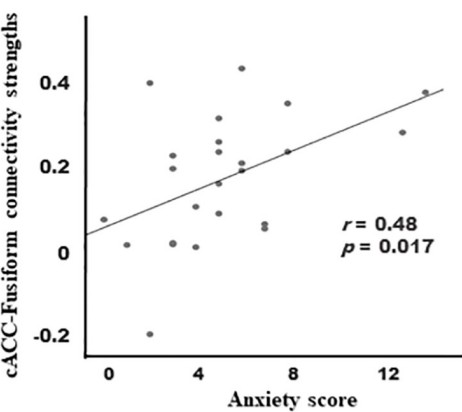

**Fig 4. The psychophysiological interaction effect showing functional connections of caudal anterior cingulate cortex (cACC) activity with bilateral fusiform gyrus activity for the father-positive condition compared to the mother-positive condition.** The connectivity strengths between the two regions showed positive correlation with the anxiety scores.

Fig 4 shows the results from PPI analysis, in which the seeds were located in clusters showing an interaction effect. Significantly increased functional connections with the right caudal ACC were found in the bilateral fusiform gyrus and postcentral gyrus for the father-positive condition compared to the mother-positive condition. No significant connections were found with the right precuneus. Correlation analysis between the connectivity strengths and psychological scale scores, the only significant result was positive correlation between the caudal ACC-fusiform gyrus connectivity strengths and anxiety scores ($r = 0.48$, $p = 0.017$).

## Discussion

The present study examined the neural relationship between the perception of a parent and the valence of an autobiographical memory, then explored how it is modulated by one's psychological health. We measured perception by inciting imagination of the facial expression a parent had made in a given situation. The three seconds given for the task may have been too short to remember an autobiographical memory. However, since the vividness and reaction intensity in the stimulus assessments were rated as moderately or higher, we can say that the task was to evoke an autobiographical memory rather than an imagination or a simple impression. From our behavioral data, we saw a rank in imagined emotion scores among the valences of childhood memories, where positive situations elicited the highest imagined emotion score, followed by neutral situations and negative situations. In conjunction with the RT results, we could infer a sense of hesitation in choosing an appropriate expression especially for negative childhood situations compared to positive situations.

The quality of communication between parents and children is an excellent indicator of the quality of a parent-child relationship. Studies suggest that open communication with parents allows children to learn skills that are closely related to healthy coping skills and problem-solving strategies [54,55]. Thus, we assessed whether the quality of the relationship with parents affects the impression of parents during childhood. As expected, we found that more open communication with either father or mother was correlated with a greater emotion score in positive emotion situations for both parents. On the other hand, negative correlations were seen between communication problem scale scores (father/mother) and the imagined emotion ratings for the respective parent-positive conditions. Altogether, these findings convey that relationship quality directly affects the recollection of one's parents in positive contexts, but

not necessarily in negative situations. Moreover, the results also revealed that one's level of self-esteem positively influences the perception of one's father, particularly in positive situations. This finding further highlighted the influence of self-esteem on how positively one perceives his or her father for positive memories, rather than rumination on exaggerated negative affects seen in previous patient studies [19,41].

From the neuroimaging data, we saw intriguing neural responses to each condition from the right caudal ACC and precuneus, in particular. In the result of the caudal ACC, there was a particular bias toward father-negative and mother-positive, suggesting that in the autobiographical memory of emotional events, different forms of brain reactions occur depending on the parent's gender. Attention on the caudal ACC has risen in recent years and studies have indicated that the region contributes significantly to negative affect and pain processing [56–58]. This region receives an ample amount of emotion-related and voluntary movement signals [59,60], and has been suggested as a pivotal site for the emotional and voluntary interaction [58]. In a fascinating investigation on pain processing in self and others, it was seen that the caudal ACC was activated in response to self-experienced pain, but not when perceiving pain in others [56]. Furthermore, this region was shown to increase in response to negatively charged autobiographical memories [61]. On the other hand, its involvement in other-experienced pain processing has also been reported. A meta-analysis on empathy-related regions indicates that this region is part of the pain empathy network, and is engaged when pain was anticipated [62]. Regardless, it is clear that the caudal ACC plays a role in processing negative affect.

In light of these caudal ACC functions, our results may be linked to mothers taking on the role of primary caretaker more so than the fathers. In many cultures, traditional gender roles have determined the roles and expectations for parents, which usually assumes fathers as breadwinners and mothers as the primary caretaker in child rearing [63,64]. This point can be related to our results of the stimulus assessment. The analysis of the stimulus assessments indicated that the responses fathers gave were less intense in all situations than those given by mothers; and the imagination of fathers' appearances were less vivid and evoked less arousal than that of the mothers. In these respects, it is possible that caudal ACC activity may be a response to the stoic and emotionally distant perception of fathers. Meanwhile, the importance of memory to the mother can be seen in our result of negative correlation between caudal ACC activity in the mother-positive condition and depression score. There was a report that pathological changes in this region may be related to major depressive disorder [65]. A study on the behavioral adaptation to unpleasant environment showed the involvement of the caudal ACC to be critical in the ability to regulate the behavior [66]. It has been suggested that the region might contribute to defending one from developing depression [67]. Taken together, our result of the negative correlation in the mother-positive condition may suggest that through the function of the caudal ACC, autobiographical memories of positive events with the mother may act as a factor that prevents depression.

The precuneus, a central hub of the default mode network, also exhibited a similar interaction effect between the parent and situation during the childhood memory recollection task. The precuneus has been thought to play a critical role in episodic memory [68,69], source monitoring [70], and self-referential processing [71]. Recently, a multivariate analysis method, representational similarity analysis, demonstrated that the precuneus is a key region engaged during the subjective experience of vivid autobiographical memory reminiscence [72]. Other experiments have confirmed its prominent involvement in the vividness of self-related memories [73,74]. In addition, right lateralization of the precuneus has been shown for highly personal and vivid reminiscence of autobiographical events [72]. Right lateralization is corroborated by the present study, which required participants not only to visualize a given circumstance, but also to delineate the facial expression by either

parent. Our results indicate overall greater activation of the right precuneus during recollection of fathers than mothers, and more specifically, its activity was higher when recollecting fathers in negative compared to positive situations. It is interesting to note that despite lower vividness score in the stimulus assessments for fathers than mothers, these findings were observed, suggesting that the perceptions of fathers may be more personal than those of mothers, especially during reminiscence of negative situations. These may indicate the saliency of childhood memories of fathers and particularly how fathers responded during negative circumstances. In conjunction with activity of the caudal ACC, we cautiously conclude that memories of fathers in negative situations have a more adverse effect as the negative impression of fathers is more permanently embedded. This interpretation can also be linked to the results of correlation with self-esteem.

Studies report that the neural structure of personal semantic memories serves as a neural correlate of self-esteem [75,76]. It is postulated that due to its role in using autobiographical memories to choose the applicable self-relevant information, activation of the precuneus is closely related to self-esteem and how one processes given information [76,77]. The current study confirmed the established association between self-esteem and precuneus activity, as we found a negative relationship between one's self-esteem and precuneus activity during autobiographical reminiscence of fathers in negative events. Recently, the precuneus was identified to correlate positively with evaluation sensitivity, measured by a change in state self-esteem [75]. Together with the evidence supporting the precuneus' affiliation with evaluation sensitivity, as well as the tendency for individuals with lower self-esteem to be more affected by negative information [76], our result seems to suggest that the heightened activation observed in individuals with lower self-esteem may be due to the saliency of negative memories during recollection and their tendency to be more influenced by negative information.

Our PPI analysis revealed significant increases in functional connectivity of the caudal ACC with the fusiform gyrus and postcentral gyrus for the father-positive condition compared to the mother-positive condition. Given that the fusiform gyrus processes facial identity [78] and emotion [79], our finding may indicate that a greater sense of negative affect was involved while processing facial expressions of fathers than mothers in positive childhood events. This raises the possibility that the negative affect arisen from processing emotional face expressions may be particularly different between the parent in a positive context. Going back to the stimulus assessments, we saw lower vividness, response intensity, and arousal from fathers than mothers. Again, this may be since, generally, mothers are primary caregivers who spend more time with children; thus, one may have more readily available positive memories involving mothers than fathers to access on demand.

Previously, individuals with high trait anxiety have shown to hyperactivate the fusiform gyrus while processing uncertain cues and angry faces [80,81]. Consistent with the activation study, our study revealed that the connectivity strengths between the cingulate cortex and fusiform gyrus during the father-positive versus mother-positive condition correlated positively with the severity of anxiety symptoms. These results may provide the neurobiological influence of trait anxiety in the retrieval of memory and the perception of fathers. In addition to emotional face processing, greater connectivity between the cingulate area and postcentral gyrus was witnessed for the father-positive than mother-positive condition. Functional connectivity between these two regions has been implicated in posttraumatic stress disorder (PTSD), where patients with PTSD showed stronger connectivity than the control group, and also was positively correlated with the severity of symptoms [82]. As part of the sensorimotor network [83], the coupling of the caudal ACC and postcentral or paracentral gyri has been shown to be involved in shock processing [84] and other emotional memory tasks [85]. Based on the

evidence, we postulate that the caudal ACC may relay negatively valenced affective information to the sensorimotor area more for fathers than mothers when processing facial expressions.

There are some potential limitations in the current study. First, in the relationship with parents, the degree of intimacy or memory biases can vary between father and mother, depending on the individual's sex. Because of the small sample size, however, the effect of sex variation was not analyzed. Second, since the correlation analysis was performed between three scale scores and regional activity under each condition, the possibility of false positives should be considered in interpreting the results. Third, as the present study required participants to engage in an autobiographical memory retrieval task regarding their childhood, it would have been interesting to investigate the quality of their parent-child relationship their pre-adulthood period. Fourth, it also would have been helpful to acquire current living situation data as research indicates some relationships between child and parent change depending on the child's living situation [86]. Similarly, the current study did not survey parental divorce as it has been shown to influence various aspects of childhood development [87]. However, all participants in the current study reported having had relationships with both parents in childhood, regardless of parental divorce status. Nevertheless, in future studies, incorporating the status of parental divorce, quality of parent-child relationship in childhood in the study to examine various effects of divorce, age, sex, etc. are imperative to achieve a more comprehensive understanding of the perception of parents during autobiographical memory reminiscence.

In conclusion, the present study sheds light on neurobiological differences in the reminiscence of parents and situational valence. The caudal ACC and precuneus responded differently depending on which parent was the target as well as the valence of the situation in which participants were to recollect. Interestingly, both neural regions exhibited similar behavior, where activations were heightened for the father-negative and mother-positive conditions. Activities of the caudal ACC and precuneus negatively correlated with the severity of depressive symptoms as well as the level of self-esteem, respectively. Together, the negative relationships highlight the influence of depression severity and self-esteem on the negative affect processing and the personalness of memories. In addition, task-dependent functional connectivity between the caudal ACC and fusiform gyrus was greater for the father-positive than mother-positive condition. The positive correlation between anxiety level and the strength of connectivity between the caudal ACC and fusiform gyrus highlights the consequence of anxiety on negative emotions arisen from face processing. The findings successfully demonstrate the modulatory effect of psychological health on the perception of parents during autobiographical reminiscence.

## Supporting information

**S1 Table. The list of situations presented in the experiment.**
(DOCX)

## Author Contributions

**Conceptualization:** Eun Seong Kim, Jae-Jin Kim.

**Data curation:** Hesun Erin Kim.

**Formal analysis:** Eun Seong Kim.

**Funding acquisition:** Jae-Jin Kim.

**Investigation:** Eun Seong Kim, Hesun Erin Kim.

**Methodology:** Jae-Jin Kim.

**Supervision:** Jae-Jin Kim.

**Writing – original draft:** Eun Seong Kim, Hesun Erin Kim.

**Writing – review & editing:** Jae-Jin Kim.

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
