## [Decision Letter · Decision Letter 0]

18 Feb 2020

PONE-D-19-30869

The neural influence of autobiographical memory related to the parent-child relationship on psychological health in adulthood

PLOS ONE

Dear Authors,

Thank you for submitting your manuscript to PLOS ONE. After careful consideration, we feel that it has merit but does not fully meet PLOS ONE’s publication criteria as it currently stands. Therefore, we invite you to submit a revised version of the manuscript that addresses the points raised during the review process.

We would appreciate receiving your revised manuscript by 3/17/20. To enhance the reproducibility of your results, we recommend that if applicable you deposit your laboratory protocols in protocols.io, where a protocol can be assigned its own identifier (DOI) such that it can be cited independently in the future. For instructions see: http://journals.plos.org/plosone/s/submission-guidelines#loc-laboratory-protocols

We look forward to receiving your revised manuscript.

Kind regards,

Luca Cerniglia, PhD

Academic Editor

PLOS ONE

Additional Editor Comments (if provided):

For this manuscript one referee recommended rejections, another one recommended major revisions. In light of the fact that also the reviewer who recommended rejection found merit in the paper, my decision is to ask the authors to revise the article with major revision.

Journal Requirements:

3. We note that Figure 1 includes an image of a participant in the study. 

Reviewers' comments:

Reviewer's Responses to Questions

**Comments to the Author**

1. Is the manuscript technically sound, and do the data support the conclusions?

Reviewer #1: No

Reviewer #2: Partly

2. Has the statistical analysis been performed appropriately and rigorously? 

Reviewer #1: I Don't Know

Reviewer #2: Yes

3. Have the authors made all data underlying the findings in their manuscript fully available?

Reviewer #1: No

Reviewer #2: Yes

4. Is the manuscript presented in an intelligible fashion and written in standard English?

Reviewer #1: Yes

Reviewer #2: No

5. Review Comments to the Author

Reviewer #1: Review title: The neural influence of autobiographical memory related to the parent-child relationship on psychological health in adulthood

The current paper investigates the brain regions associated with the recall of autobiographical memories of peoples’ childhood memories of parental facial expressions in certain childhood situations. Specifically, looking into the association between current psychological health (self-esteem, anxiety, and depression), quality of parent-child-relationship, and specific brain regions activities.

Overall: The paper is a very interesting study and there is potential for novel information. The biggest criticisms are the characterization of the task as an autobiographical memory retrieval task, the lack of clear theory, and the major reverse inferencing due to a lack of supporting measures. Additional data is needed to support any of the claims the authors make.

Introduction: The introduction needs a rewrite. The paper is bit hard to follow. The authors in a single paragraph talk about several topics, e.g., emotion regulation, global-self-esteem, anxiety, depression, memory biases, etc. Each topic should be within its own paragraph/section. One section can be on self-esteem and memory, anxiety and memory, etc.

Also, explaining more in-depth of the specific brain regions associated with the recall of autobiographical memories, emotional memories, since this is the specific task participants will be engaging in.

Also, the hypothesis of this paper are also not clear. For example, “We predicted that emotional ratings would be correlated with psychological health facets.” What emotional ratings are being referenced? Rating of the specific parental facial expressions expressed by a parent during a given encounter or the valence of the memory in general. Furthermore, there was no justification for why they hypothesized different activity in memory regions based on parent or emotion.

Methods: A concern of the study is whether participants actually engaged in autobiographical memory recall rather than imagination or simply an impression. First, there is substantial research showing that retrieving specific autobiographical memories takes more time than 3 seconds in response to a cue word (see Conway, 2005)! In addition, participants were not actually instructed to retrieve a memory, but rather to imagine their parents’ facial expression. This type of task will be heavily influenced by current/recent experiences with their parents’ facial expression. Lastly, I’m not sure it is possible to both retrieve a memory and make the facial judgment within 3 seconds. Because participants were forced to respond so quickly (and did so looking at the RT’s), this task is best interpreted as an impression task. Do the authors have any evidence to support their interpretation of the task?

The authors also report using a linear mixed model to analyze the behavior. First, this model is quite complex and much more information is needed regarding how the model was set up (first level, second level factors, etc) and why this model was chosen over an ANOVA.

Results: The authors should report all results and all p-values e.g. …”(t = 2.77, p < 0.05)” (p. 9); “but showed no parent effect and a significant parents × situation interaction.” (p. 8) Can the authors also show figures of the main effects? The authors also do not appropriately describe the interactions (which are based on differences between conditions). The authors’ worded the interaction as main effects (“ACC activity was significantly greater in the father condition than in the mother condition”), which were inaccurate.

The authors should reconsider the names of their regions and interpretation. The ACC activity they found is better characterized as mid cingulate cortex. It is a bit too posterior to be considered dACC activity.

Motivation for the correlations do not make sense. For example, the authors found activity in the “ACC” that was driven by valence differences in the father condition, and yet found a correlation between depression and brain activity in this region in the mother-positive condition! Same with the other correlation. This implies that the authors assessed all possible correlations, regardless as to whether they made sense which would be at least 24 tests (3 scales, 4 conditions, 2 regions), thus inflating false positives. Not only should these correlations not be trusted (due to false positives), but they also were not hypothesized.

Discussion: The major issue in the discussion is that the authors over-interpret their findings and use a lot of reverse inferencing to do so. As one example, the authors claim that the dACC activity represents conflict processing. First, I am not convinced that the dACC was activated at all. However, even if this were to be the dACC, the dACC is one of the most active areas and does not indicate that conflict processing has occurred. The authors would benefit from reading Poldrack, R. A. (2011). Inferring mental states from neuroimaging data: from reverse inference to large-scale decoding. Neuron, 72(5), 692-697. Another example is that the authors claim that they can “therefore infer that the perceptions of fathers were more personal and vivid than those of mothers, especially during reminiscence of negative situations.” Again, none of this can actually be inferred without actual ratings of vividness or “personalness”. Because the Discussion is primarily comprised of entirely reverse inferencing and over-interpretation, it would have to be rewritten and I am convinced that nothing can be firmly concluded without additional data.

One source of evidence of over-interpretation is again the interpretation of conflict monitoring. Decades of research has shown that conflict comes with a cost in response times. And yet, this “conflict monitoring” that occurs only for fathers and not mothers is not borne out in the RT ratings.

Lastly, the authors do indicate one of the major limitations is not being able to analyze sex differences due to the small sample size (which I agree with), but the authors may not realize how much of a problem this actually is. Because the authors have already chosen to differentially investigate effect mother vs. father effects, the relationship and memory biases that one might have will surely depend on the sex of the individual. The net result is an average set of brain activity that may not represent either the men or women (e.g., the average of 0 and 10 is 5 and 5 is equally as far from representing men and women, nor represent either).

Recommendations: In the end, I would highly recommend testing a new set of participants behaviorally in the same task, but with additional probing to test whether they actually retrieved a memory, to rate the vividness of that memory, assess the degree of conflict they experienced, and any other rating to support the inferences the authors are making. Making claims of cognitive processing based on the neuroimaging findings alone without such support is a critical and known inferencing flaw in the neuroimaging community.

Reviewer #2: Thank you very much for the possibility to review the manuscript titled "The neural influence of autobiographical memory related to the parent-child relationship on psychological health in adulthood". I think that this study contributes a lot to the literature and should be published in this journal if the authors want to review some aspects of the article that are lacking. This study investigates the neural influence on autobiographical memories; however, the introductory section refers to several concepts not explored in this work: emotional regulation, parent-child attachment. I think it may be important to review the whole introductory section in order to better focus the present work. In fact, the authors are invited to review studies related to the parent-child relationship (rather than to the model of indivisual attachment) and to deepen the literature on autobiographical memory.

For example, the authors are recommended to deepen their knowledge of the following works, which would greatly enrich this manuscript:

Fivush, R., & Haden, C. A. (Eds.). (2003). Autobiographical memory and the construction of a narrative self: Developmental and cultural perspectives. Psychology Press.

Fivush, R. et al. (1996). Remembering, recounting, and reminiscing: The development of autobiographical memory in social context.

Cimino, S. et al. (2020). Dialogues about Emotional Events between Mothers with Anxiety, Depression, Anorexia Nervosa, and No Diagnosis and Their Children. Parenting, 20(1), 69-82.

Again with regard to the introductory section, the authors are invited to define more clearly the objectives of the work and the hypotheses, consistent with the results and discussions.

Discussions can also be reviewed in the light of the cited literature and authors are invited to use an editing service for English language.

6. PLOS authors have the option to publish the peer review history of their article (what does this mean?). If published, this will include your full peer review and any attached files.

Reviewer #1: No

Reviewer #2: No

---

## [Author Response · Author response to Decision Letter 0]

18 Mar 2020

Reviewer #1: 

Review title: The neural influence of autobiographical memory related to the parent-child relationship on psychological health in adulthood

The current paper investigates the brain regions associated with the recall of autobiographical memories of peoples’ childhood memories of parental facial expressions in certain childhood situations. Specifically, looking into the association between current psychological health (self-esteem, anxiety, and depression), quality of parent-child-relationship, and specific brain regions activities.

Overall: The paper is a very interesting study and there is potential for novel information. The biggest criticisms are the characterization of the task as an autobiographical memory retrieval task, the lack of clear theory, and the major reverse inferencing due to a lack of supporting measures. Additional data is needed to support any of the claims the authors make.

Reply:

We greatly appreciate the positive evaluation and valuable comments on the manuscript revision. Problems pointed out were all reflected in the revision of the manuscript.

Introduction: The introduction needs a rewrite. The paper is bit hard to follow. The authors in a single paragraph talk about several topics, e.g., emotion regulation, global-self-esteem, anxiety, depression, memory biases, etc. Each topic should be within its own paragraph/section. One section can be on self-esteem and memory, anxiety and memory, etc.

Reply:

As suggested, the introduction was totally rewritten. Autobiographical memory, memory biases, self-esteem, depression, and anxiety were described as separate paragraphs and contents were supplemented. The emotion regulation-related content was deleted because it did not fit the subject of this paper. In addition, we added some additional background on the nature of parental roles and how processing fathers and mothers engage the brain differently.

Revision:

In ‘the first to 7th paragraphs’ of the Introduction section

Like the saying, “no health without mental health,” promoted by the WHO, psychological health or psychological well-being is a fundamental factor contributing to quality of life [1]. There are many factors that influence psychological health in adulthood, and one of the most central determinants is the parent-child relationship [2]. The relationship between a parent and child becomes the template for other interpersonal relationships because attachment bonding allows a person to understand their environment and establish a sense of security [3,4]. An examination of the association between the retrospective perceptions of the parent-child relationship and emotional health in adulthood showed that daily emotional and psychological distress are closely related to how one perceives the quality of the parent-child relationship, and the retrospective perceptions of father-child and mother-child relationships contribute to different dimensions of emotional health [5]. Therefore, the perceptions of the childhood relationship with parents are important in the overall well-being, even as adults.

The retrospective perceptions of the parent-child relationship are based on an individual’s autobiographical memory. The development of autobiographical memory is a social process that is influenced by the reminiscing style of parents, and thus an integral part of the developing sense of self and interpersonal relatedness [6,7]. Autobiographical memory represents the self and shapes emotional states of our everyday lives [8,9]. In general, memories are highly intertwined with features like self-referential processing and emotion [9]. Functional neuroimaging studies have examined autobiographical memories in reference to other types of memories. For instance, a recent study that compared the difference between autobiographical memory and simple recognition memory found that when a task required retrieval of life memories, default-mode areas were engaged more, whereas a simple recollection of memories engaged the parietal memory network [10]. Since autobiographical memory retrieval relies heavily on self-reference and emotion, retrieval of autobiographical memories has been shown to employ various cortical and subcortical regions, including the prefrontal and parietal cortices, anterior cingulate cortex (ACC), precuneus, medial temporal cortex, hippocampus, and amygdala [11-14]. In particular, the amygdala has been implicated in the encoding of emotional memories [15] as well as retrieval of autobiographical memories [16]. Previous studies also found coactivation of the amygdala, hippocampus, and inferior frontal gyrus (IFG) during the retrieval of autobiographical memories [17] and neural responses in the dorsal to caudal ACC and IFG elicited by the retrieval of painful autobiographical memories [18]. These reports consistently show a close relationship between autobiographical memory and emotional memory.

Psychological health is very closely related to memory encoding and retrieval, and there is evidence supporting its influence on neural activities related to autobiographical memory. Numerous studies have also shown the connection between psychological health and selective memory bias. The representation, recall, and maintenance of autobiographical memories have shown to be disturbed by individuals with poor psychological well-being, where they are systematically biased for negative information [11]. Memory bias and negative interpretations of events contribute to a vicious cycle, where one’s psychological health contributes to the tendency toward negative interpretation of memories, and this bias also maintains low self-esteem, depression, or anxiety [19-21]. In fact, there is a consensus that self-esteem, anxiety, and depression are core elements of psychological health [22-24]. The importance of these elements has spurred researchers to study the close associations between self-esteem, anxiety and depression, and quality of life in numerous fields [25-28].

Self-esteem is a facet of personality that affects perception of social standing and modulates the salience of social interaction. Previous studies have shown that the midline cortical structures, such as the medial prefrontal cortex (PFC), ACC, posterior cingulate cortex (PCC), and precuneus that mediate self-referential processing are also closely involved in the expression of self-esteem [29]. Global self-esteem, which includes self-competence and self-liking, is related to selective memory, especially toward a negatively salient stimulus [19]. Individuals with lower self-esteem tend to be more concerned with how others perceive them, and this makes heightened negative events more memorable. Lower self-esteem predicts an increase in ventral ACC and medial PFC activity in response to positive versus negative social feedback, indicating the relationship between self-esteem and the salience processing [30]. On the other hand, a positivity bias for events that involve self-evaluations is stronger for individuals with higher self-esteem, suggesting that this bias affecting autobiographical memory is part of psychological mechanisms for maintaining a positive self-image [31]. The underlying neural substrates of a positive self-evaluation include the medial ventral and dorsolateral PFC and hippocampus, reflecting cognitive effort and emotional involvement for the positivity bias [32]. In addition, it has been reported that the lateral PFC, dorsal ACC, posterior cingulate cortex (PCC), precuneus, and caudate are involved in negative or positive character feedback associated with self-esteem [33].

The level of self-esteem may be related to the manifestation of depression through memory biases [34]. Many studies have reported that overall autobiographical memory predicts the onset or course of depression and thus is considered to be a risk factor for depression [35,36]. Depression contributes to the deviant reactivation of neural areas associated with autobiographical memory retrieval [37]. Neural correlates of autobiographical memory deficits in depressed patients have been known to involve a wide range of brain regions, such as the IFG, lateral orbitofrontal cortex (OFC), dorsomedial and ventrolateral PFC, ACC, precuneus, temporoparietal cortex, insula, hippocampus, and amygdala [38-40]. 

Pathological anxiety has also been associated with negative memory bias [41]. Like depression, anxiety disorders are also characterized by maladaptive emotional responses and retrieval of memories [42]. In particular, negative autobiographical memory bias is prominent in individuals high in social anxiety [43]. Increased state anxiety decreases the ability to retrieve specific autobiographical memory [44]. Previous studies of patients with anxiety disorders have reported functional impairments of structures linked to the experience and regulation of emotion, such as the PFC, ACC, insula, and amygdala [45-47].

Although behavioral and neurobiological studies indicate the impact of the psychological dimensions on the encoding and retrieval of autobiographical memories, the way in which psychological health affects memory processes in the general population is still unknown. The importance of the parent-child relationship in child development has been well documented. When looking at the way parents interact with a child, mothers are often more caring and nurturing, whereas fathers engage in more playful interactions [48]. Naturally, since the relationship between a father and child is different from the relationship between a mother and child, the autobiographical memories of parents in adulthood must be different. This difference is also observed in the brain response. For example, faces of mothers elicited more activity in the face processing network including the fusiform gyrus, whereas faces of fathers elicited more activity in the striatal region [49]. Despite the importance of the parent-child relationship and its difference between parents, no study has provided neurobiological evidence for how perceptions change depending on the parent and the valence of memories.

Also, explaining more in-depth of the specific brain regions associated with the recall of autobiographical memories, emotional memories, since this is the specific task participants will be engaging in.

Reply:

As suggested, additional explanations regarding autobiographical memory and emotional memory were added to a separate paragraph.

Revision:

In ‘the second paragraph’ of the Introduction section

The retrospective perceptions of the parent-child relationship are based on an individual’s autobiographical memory. The development of autobiographical memory is a social process that is influenced by the reminiscing style of parents, and thus an integral part of the developing sense of self and interpersonal relatedness [6,7]. Autobiographical memory represents the self and shapes emotional states of our everyday lives [8,9]. In general, memories are highly intertwined with features like self-referential processing and emotion [9]. Functional neuroimaging studies have examined autobiographical memories in reference to other types of memories. For instance, a recent study that compared the difference between autobiographical memory and simple recognition memory found that when a task required retrieval of life memories, default-mode areas were engaged more, whereas a simple recollection of memories engaged the parietal memory network [10]. Since autobiographical memory retrieval relies heavily on self-reference and emotion, retrieval of autobiographical memories has been shown to employ various cortical and subcortical regions, including the prefrontal and parietal cortices, anterior cingulate cortex (ACC), precuneus, medial temporal cortex, hippocampus, and amygdala [11-14]. In particular, the amygdala has been implicated in the encoding of emotional memories [15] as well as retrieval of autobiographical memories [16]. Previous studies also found coactivation of the amygdala, hippocampus, and inferior frontal gyrus (IFG) during the retrieval of autobiographical memories [17] and neural responses in the dorsal to caudal ACC and IFG elicited by the retrieval of painful autobiographical memories [18]. These reports consistently show a close relationship between autobiographical memory and emotional memory.

Also, the hypothesis of this paper are also not clear. For example, “We predicted that emotional ratings would be correlated with psychological health facets.” What emotional ratings are being referenced? Rating of the specific parental facial expressions expressed by a parent during a given encounter or the valence of the memory in general. Furthermore, there was no justification for why they hypothesized different activity in memory regions based on parent or emotion.

Reply:

We sincerely apologize for the confusion. To answer the first part, the “emotional ratings” we mentioned in the hypothesis was the in-scan behavior data. In the revised manuscript, the hypothesis was updated to avoid the confusion.

Revision:

In ‘the last paragraph’ of the Introduction section

In this study of the general population, we sought to identify the neural substrates of the impression of parents while participants recalled childhood memories. For this purpose, we examined how fathers and mothers are perceived differently and whether the valence of memory influences this perception, and surveyed the effects of psychological health (self-esteem, depression, and anxiety) on the memory process. As childhood memories of parents triggered by various situations are inherently charged with emotions, we hypothesized that regions often involved in an autobiographical memory, such as the ACC, IFG, amygdala, and hippocampus, would be engaged differently depending on the parent and the valence of situation, and would be modulated by a person’s psychological health.

Methods: A concern of the study is whether participants actually engaged in autobiographical memory recall rather than imagination or simply an impression. First, there is substantial research showing that retrieving specific autobiographical memories takes more time than 3 seconds in response to a cue word (see Conway, 2005)! In addition, participants were not actually instructed to retrieve a memory, but rather to imagine their parents’ facial expression. This type of task will be heavily influenced by current/recent experiences with their parents’ facial expression. Lastly, I’m not sure it is possible to both retrieve a memory and make the facial judgment within 3 seconds. Because participants were forced to respond so quickly (and did so looking at the RT’s), this task is best interpreted as an impression task. Do the authors have any evidence to support their interpretation of the task?

Reply:

We are sincerely grateful for this particular point. As indicated, 3 seconds can actually be a very short time to remember an autobiographical memory. Our manuscript had no basis to properly answer this pointed problem. Now, following your recommendations at the end, we provided clarity data for task stimuli after the additional experiment. As shown in the newly added Table 1, the vividness of memory associated with the situation and the intensity of reaction received from a parent in all situations indicated 4 or greater, which means ‘above moderately.’ We think these results may explain to an extent to what you have expressed concerns. In the revised manuscript, we added this comment in the discussion.

Revision:

In ‘the first paragraph’ of the Discussion section

The three seconds given for the task may have been too short to remember an autobiographical memory. However, since the vividness and reaction intensity in the stimulus assessments were rated as moderately or higher, we can say that the task was to evoke an autobiographical memory rather than an imagination or a simple impression.

The authors also report using a linear mixed model to analyze the behavior. First, this model is quite complex and much more information is needed regarding how the model was set up (first level, second level factors, etc) and why this model was chosen over an ANOVA.

Reply:

As pointed out, it seems that LMM was not necessary, and thus ANOVA was applied to the revised manuscript. As the analysis method changed, it was adjusted accordingly to the statistical values of the results.

Revision:

In ‘Behavioral task and analysis’ of the Methods section

The ANOVA was used to identify the main effect of parents (father and mother) and situation (positive, negative, and neutral) and the interaction effect between them.

In ‘Behavioral results’ of the RESULTS section

Figure 2 illustrates the mean imagined emotion score and mean RT in each condition during the task. The imagined emotion scores showed a main effect of situation [F2,47 = 408.28, p < 0.001], but not a main effect of parents [F1,71 = 0.45, p = 0.504] and a significant parents × situation interaction [F2,23 = 0.80, p = 0.460]. Post-hoc analyses showed that the scores for the positive situations were significantly higher than those for the negative (t = 20.49, p < 0.01) and neutral situations (t = 14.53, p < 0.01), and the scores for the negative situations were significantly lower than those for the neutral situations (t = -15.86, p < 0.01).

The RTs also showed a main effect of situation [F2,47 = 45.42, p < 0.001] and a significant parents × situation interaction [F2,23 = 4.42, p = 0.024], but not a main effect of parents [F1,71 = 0.06, p = 0.813]. Post-hoc analyses showed that the RTs for the positive situations were significantly shorter than those for the negative situations (t = -6.82, p < 0.01) and the neutral situations (t = -6.77, p < 0.01), and there was no significant difference between the negative and neutral situations (t = 1.49, p = 0.151). The RTs in the father-negative condition were significantly longer than those in the father-neutral condition (t = 3.90, p < 0.01), whereas there was no significant difference between the mother-negative and mother-neutral conditions (t = 1.49, p = 0.453).

Results: The authors should report all results and all p-values e.g. …”(t = 2.77, p < 0.05)” (p. 9); “but showed no parent effect and a significant parents × situation interaction.” (p. 8) Can the authors also show figures of the main effects? The authors also do not appropriately describe the interactions (which are based on differences between conditions). The authors’ worded the interaction as main effects (“ACC activity was significantly greater in the father condition than in the mother condition”), which were inaccurate.

Reply:

As suggested, we added statistics for the results without significance. Since the differences among the positive, negative, and neutral conditions were so similar between the father and mother conditions, we did not add a graph of the main effect to avoid the duplication. In terms of the interaction effect, we agree with the comment that our reports were inaccurate. In the revised results, we changed them into an appropriate form.

Revision:

In ‘Behavioral results’ of the RESULTS section

Figure 2 illustrates the mean imagined emotion score and mean RT in each condition during the task. The imagined emotion scores showed a main effect of situation [F2,47 = 408.28, p < 0.001], but not a main effect of parents [F1,71 = 0.45, p = 0.504] and a significant parents × situation interaction [F2,23 = 0.80, p = 0.460]. Post-hoc analyses showed that the scores for the positive situations were significantly higher than those for the negative (t = 20.49, p < 0.01) and neutral situations (t = 14.53, p < 0.01), and the scores for the negative situations were significantly lower than those for the neutral situations (t = -15.86, p < 0.01).

The RTs also showed a main effect of situation [F2,47 = 45.42, p < 0.001] and a significant parents × situation interaction [F2,23 = 4.42, p = 0.024], but not a main effect of parents [F1,71 = 0.06, p = 0.813]. Post-hoc analyses showed that the RTs for the positive situations were significantly shorter than those for the negative situations (t = -6.82, p < 0.01) and the neutral situations (t = -6.77, p < 0.01), and there was no significant difference between the negative and neutral situations (t = 1.49, p = 0.151). The RTs in the father-negative condition were significantly longer than those in the father-neutral condition (t = 3.90, p < 0.01), whereas there was no significant difference between the mother-negative and mother-neutral conditions (t = 1.49, p = 0.453).

In ‘Imaging results’ of the RESULTS section

There was an interaction effect in the right caudal ACC and right precuneus. As shown in Figure 3A, right caudal ACC activity in the father condition was significantly greater in the negative situation than in the positive situation (t = 5.09, p < 0.01), whereas that in the mother condition was not significant but greater in the positive condition than in the negative condition (t = 1.34, p = 0.193). Figure 3B also shows that right precuneus activity in the father condition was significantly greater in the negative situation than in the positive situation (t = 2.35, p < 0.05), whereas that in the mother condition was not significant but greater in the positive condition than in the negative condition (t = 1.94, p = 0.065).

The authors should reconsider the names of their regions and interpretation. The ACC activity they found is better characterized as mid cingulate cortex. It is a bit too posterior to be considered dACC activity.

Reply:

We agree with this comment. I have two significant results in the ACC. In the revised manuscript, the first one in the main effect of situation was named the dorsal ACC, and the second one in the interaction effect was regarded as the caudal ACC. As a result of this separation, the interpretation in the discussion was changed accordingly.

Revision:

In ‘Imaging results’ of the Results section

There was an interaction effect in the right caudal ACC and right precuneus.

In ‘3rd and 4th paragraphs’ of the Discussion section

From the neuroimaging data, we saw intriguing neural responses to each condition from the right caudal ACC and precuneus, in particular. In the result of the caudal ACC, there was a particular bias toward father-negative and mother-positive, suggesting that in the autobiographical memory of emotional events, different forms of brain reactions occur depending on the parent's gender. Attention on the caudal ACC has risen in recent years and studies have indicated that the region contributes significantly to negative affect and pain processing [56-58]. This region receives an ample amount of emotion-related and voluntary movement signals [59,60], and has been suggested as a pivotal site for the emotional and voluntary interaction [58]. In a fascinating investigation on pain processing in self and others, it was seen that the caudal ACC was activated in response to self-experienced pain, but not when perceiving pain in others [56]. Furthermore, this region was shown to increase in response to negatively charged autobiographical memories [61]. On the other hand, its involvement in other-experienced pain processing has also been reported. A meta-analysis on empathy-related regions indicates that this region is part of the pain empathy network, and is engaged when pain was anticipated [62]. Regardless, it is clear that the caudal ACC plays a role in processing negative affect.

In light of these caudal ACC functions, our results may be linked to mothers taking on the role of primary caretaker more so than the fathers. In many cultures, traditional gender roles have determined the roles and expectations for parents, which usually assumes fathers as breadwinners and mothers as the primary caretaker in child rearing [63,64]. This point can be related to our results of the stimulus assessment. The analysis of the stimulus assessments indicated that the responses that fathers gave were less intense in all situations than those given by mothers; and the imagination of fathers’ appearances were less vivid and evoked less arousal than that of the mothers. In these respects, it is possible that caudal ACC activity may be a response to the stoic and emotionally distant perception of fathers. Meanwhile, the importance of memory to the mother can be seen in our result of negative correlation between caudal ACC activity in the mother-positive condition and depression score. There was a report that pathological changes in this region may be related to major depressive disorder [65]. A study on the behavioral adaptation to unpleasant environment showed the involvement of the caudal ACC to be critical in the ability to regulate the behavior [66]. It has been suggested that the region might contribute to defending one from developing depression [67]. Taken together, our result of the negative correlation in the mother-positive condition may suggest that through the function of the caudal ACC, autobiographical memories of positive events with the mother may act as a factor that prevents depression.

Motivation for the correlations do not make sense. For example, the authors found activity in the “ACC” that was driven by valence differences in the father condition, and yet found a correlation between depression and brain activity in this region in the mother-positive condition! Same with the other correlation. This implies that the authors assessed all possible correlations, regardless as to whether they made sense which would be at least 24 tests (3 scales, 4 conditions, 2 regions), thus inflating false positives. Not only should these correlations not be trusted (due to false positives), but they also were not hypothesized.

Reply:

As defined in the introduction, we hypothesized that a person’s psychological health (self-esteem, anxiety, and depression) would be modulated by neural regions including the ACC, IFG, hippocampus, amygdala, and other areas associated with autobiographical memory recall. Since the regions of interest (caudal ACC and precuneus) result from the interaction effect, we considered their activity meaningful in every condition. We saw that if there was a difference in the father condition, there was also a meaning of no difference in the mother condition, and vice versa. Of course, as pointed out, it is a problem that the likelihood of false positives increases. We added this problem in the limitation of the study in the discussion.

Revision:

In the 9th paragraph of the DISCUSSION section

Second, since the correlation analysis was performed between three scale scores and regional activity under each condition, the possibility of false positives should be considered in interpreting the results.

Discussion: The major issue in the discussion is that the authors over-interpret their findings and use a lot of reverse inferencing to do so. As one example, the authors claim that the dACC activity represents conflict processing. First, I am not convinced that the dACC was activated at all. However, even if this were to be the dACC, the dACC is one of the most active areas and does not indicate that conflict processing has occurred. The authors would benefit from reading Poldrack, R. A. (2011). Inferring mental states from neuroimaging data: from reverse inference to large-scale decoding. Neuron, 72(5), 692-697. Another example is that the authors claim that they can “therefore infer that the perceptions of fathers were more personal and vivid than those of mothers, especially during reminiscence of negative situations.” Again, none of this can actually be inferred without actual ratings of vividness or “personalness”. Because the Discussion is primarily comprised of entirely reverse inferencing and over-interpretation, it would have to be rewritten and I am convinced that nothing can be firmly concluded without additional data.

One source of evidence of over-interpretation is again the interpretation of conflict monitoring. Decades of research has shown that conflict comes with a cost in response times. And yet, this “conflict monitoring” that occurs only for fathers and not mothers is not borne out in the RT ratings.

Reply:

We agree with the comments that our discussion contained too much of over-interpretation. As suggested, we included additional behavioral data in order to clarify certain claims that we made. We also edited our claims to eliminate reverse inferencing and over-interpretation as much as possible. Additionally, we changed the dACC region significant from the interaction effect to the caudal ACC. Accordingly, the discussion section of the revised manuscript was rewritten entirely.

Revision:

In ‘the 3rd to 8th paragraphs’ of the Discussion section

From the neuroimaging data, we saw intriguing neural responses to each condition from the right caudal ACC and precuneus, in particular. In the result of the caudal ACC, there was a particular bias toward father-negative and mother-positive, suggesting that in the autobiographical memory of emotional events, different forms of brain reactions occur depending on the parent's gender. Attention on the caudal ACC has risen in recent years and studies have indicated that the region contributes significantly to negative affect and pain processing [56-58]. This region receives an ample amount of emotion-related and voluntary movement signals [59,60], and has been suggested as a pivotal site for the emotional and voluntary interaction [58]. In a fascinating investigation on pain processing in self and others, it was seen that the caudal ACC was activated in response to self-experienced pain, but not when perceiving pain in others [56]. Furthermore, this region was shown to increase in response to negatively charged autobiographical memories [61]. On the other hand, its involvement in other-experienced pain processing has also been reported. A meta-analysis on empathy-related regions indicates that this region is part of the pain empathy network, and is engaged when pain was anticipated [62]. Regardless, it is clear that the caudal ACC plays a role in processing negative affect.

In light of these caudal ACC functions, our results may be linked to mothers taking on the role of primary caretaker more so than the fathers. In many cultures, traditional gender roles have determined the roles and expectations for parents, which usually assumes fathers as breadwinners and mothers as the primary caretaker in child rearing [63,64]. This point can be related to our results of the stimulus assessment. The analysis of the stimulus assessments indicated that the responses that fathers gave were less intense in all situations than those given by mothers; and the imagination of fathers’ appearances were less vivid and evoked less arousal than that of the mothers. In these respects, it is possible that caudal ACC activity may be a response to the stoic and emotionally distant perception of fathers. Meanwhile, the importance of memory to the mother can be seen in our result of negative correlation between caudal ACC activity in the mother-positive condition and depression score. There was a report that pathological changes in this region may be related to major depressive disorder [65]. A study on the behavioral adaptation to unpleasant environment showed the involvement of the caudal ACC to be critical in the ability to regulate the behavior [66]. It has been suggested that the region might contribute to defending one from developing depression [67]. Taken together, our result of the negative correlation in the mother-positive condition may suggest that through the function of the caudal ACC, autobiographical memories of positive events with the mother may act as a factor that prevents depression.

The precuneus, a central hub of the default mode network, also exhibited a similar interaction effect between the parent and situation during the childhood memory recollection task. The precuneus has been thought to play a critical role in episodic memory [68,69], source monitoring [70], and self-referential processing [71]. Recently, a multivariate analysis method, representational similarity analysis, demonstrated that the precuneus is a key region engaged during the subjective experience of vivid autobiographical memory reminiscence [72]. Other experiments have confirmed its prominent involvement in the vividness of self-related memories [73,74]. In addition, right lateralization of the precuneus has been shown for highly personal and vivid reminiscence of autobiographical events [72]. Right lateralization is corroborated by the present study, which required participants not only to visualize a given circumstance, but also to delineate the facial expression by either parent. Our results indicate overall greater activation of the right precuneus during recollection of fathers than mothers, and more specifically, its activity was higher when recollecting fathers in negative compared to positive situations. It is interesting to note that despite lower vividness score in the stimulus assessments for fathers than mothers, these findings were observed, suggesting that the perceptions of fathers may be more personal than those of mothers, especially during reminiscence of negative situations. These may indicate the saliency of childhood memories of fathers and particularly how fathers responded during negative circumstances. In conjunction with activity of the caudal ACC, we cautiously conclude that memories of fathers in negative situations have a more adverse effect as the negative impression of fathers is more permanently embedded. This interpretation can also be linked to the results of correlation with self-esteem.

 Studies report that the neural structure of personal semantic memories serves as a neural correlate of self-esteem [75,76]. It is postulated that due to its role in using autobiographical memories to choose the applicable self-relevant information, activation of the precuneus is closely related to self-esteem and how one processes given information [76,77]. The current study confirmed the established association between self-esteem and precuneus activity, as we found a negative relationship between one’s self-esteem and precuneus activity during autobiographical reminiscence of fathers in negative events. Recently, the precuneus was identified to correlate positively with evaluation sensitivity, measured by a change in state self-esteem [75]. Together with the evidence supporting the precuneus’ affiliation with evaluation sensitivity, as well as the tendency for individuals with lower self-esteem to be more affected by negative information [76], our result seems to suggest that the heightened activation observed in individuals with lower self-esteem may be due to the saliency of negative memories during recollection and their tendency to be more influenced by negative information. 

 Our PPI analysis revealed significant increases in functional connectivity of the caudal ACC with the fusiform gyrus and postcentral gyrus for the father-positive condition compared to the mother-positive condition. Given that the fusiform gyrus processes facial identity [78] and emotion [79], our finding may indicate that a greater sense of negative affect was involved while processing facial expressions of fathers than mothers in positive childhood events. This raises the possibility that the negative affect arisen from processing emotional face expressions may be particularly different between the parent in a positive context. Going back to the stimulus assessments, we saw lower vividness, response intensity, and arousal from fathers than mothers. Again, this may be since, generally, mothers are primary caregivers who spend more time with children; thus, one may have more readily available positive memories involving mothers than fathers to access on demand.

Previously, individuals with high trait anxiety have shown to hyperactivate the fusiform gyrus while processing uncertain cues and angry faces [80,81]. Consistent with the activation study, our study revealed that the connectivity strengths between the cingulate cortex and fusiform gyrus during the father-positive versus mother-positive condition correlated positively with the severity of anxiety symptoms. These results may provide the neurobiological influence of trait anxiety in the retrieval of memory and the perception of fathers. In addition to emotional face processing, greater connectivity between the cingulate area and postcentral gyrus was witnessed for the father-positive than mother-positive condition. Functional connectivity between these two regions has been implicated in posttraumatic stress disorder (PTSD), where patients with PTSD showed stronger connectivity than the control group, and also was positively correlated with the severity of symptoms [82]. As part of the sensorimotor network [83], the coupling of the caudal ACC and postcentral or paracentral gyri has been shown to be involved in shock processing [84] and other emotional memory tasks [85]. Based on the evidence, we postulate that the caudal ACC may relay negatively valenced affective information to the sensorimotor area more for fathers than mothers when processing facial expressions.

Lastly, the authors do indicate one of the major limitations is not being able to analyze sex differences due to the small sample size (which I agree with), but the authors may not realize how much of a problem this actually is. Because the authors have already chosen to differentially investigate effect mother vs. father effects, the relationship and memory biases that one might have will surely depend on the sex of the individual. The net result is an average set of brain activity that may not represent either the men or women (e.g., the average of 0 and 10 is 5 and 5 is equally as far from representing men and women, nor represent either).

Reply:

We whole-heartedly agree with this comment on the importance of exploring the sex differences in this study. Because we couldn't do it because of the small number of samples, we added the reviewer's point as an additional description to the limitation of the study.

Revision:

In the 9th paragraph of the DISCUSSION section

First, in the relationship with parents, the degree of intimacy or memory biases can vary between father and mother, depending on the individual's sex. Because of the small sample size, however, the effect of sex variation was not analyzed.

Recommendations: In the end, I would highly recommend testing a new set of participants behaviorally in the same task, but with additional probing to test whether they actually retrieved a memory, to rate the vividness of that memory, assess the degree of conflict they experienced, and any other rating to support the inferences the authors are making. Making claims of cognitive processing based on the neuroimaging findings alone without such support is a critical and known inferencing flaw in the neuroimaging community.

Reply:

We greatly appreciate this suggestion on conducting additional tests. In fact, before the experiment, we had performed a preliminary test to survey the intensity of reaction given by a parent, the vividness of each stimulus, and the level of arousal to ensure the validity of the task. However, due to the small number of subjects, the results were not included in the manuscript. In response to the reviewer’s recommendation, we carried out additional experiment to increase the number of subjects, and the data has been included in the method and results to the revised manuscript. We believe the results helped to mitigate the over-interpretation and reverse-inference problem, and we thank you again for the comment.

Revision:

In ‘Task stimuli’ of the Methods section

Participants performed a task where they were asked to imagine what their parents’ facial expression looked like in a given situation during their childhood. Visual stimuli with a grey background included black words presenting a situation in the upper part and four black and white faces of the same person expressing different emotions (very happy, a little happy, very angry, and a little angry) in the lower part (Figure 1). The number of situations used in the task was 30, including ten positive (e.g., when I did errands), ten negative (e.g., when I did not do homework), and ten neutral (e.g., when I was on the phone). All actual situations used are shown in Supplementary Table S1.

 In order to evaluate the validity of the presented words, the vividness of memory associated with the situation, the intensity of reaction received from a parent in the situation, and the current degree of arousal caused by the recollection of the situation were assessed for each situation in 24 volunteers who did not participate in the fMRI experiment (27.3 ± 1.7 years old; 11 females and 13 males). Similar to the fMRI task, each situation was presented with either word “Mother” or “Father” on top of a page, then the volunteers were instructed to visualize an image of a parent during their childhood. All three items (vividness, response intensity, and arousal) were measured on a 9-point scale (0: not at all, 4: moderately, and 8: extremely). Analysis was conducted using repeated measures ANOVA for parent (Mother and Father) and valence of situation (Positive, Negative, and Neutral) and post-hoc paired t-test.

In ‘Stimulus assessment results’ of the Results section

Results from the assessment of the vividness, response intensity, and arousal of the 30 word stimuli are presented in Table 1. The vividness showed a significant result in a main effect of parent (F1,23 = 7.63, p = 0.011), but not in a main effect of situation (F2,46 = 0.46, p = 0.637) and an interaction effect (F2,46 = 1.08, p = 0.347). In post-hoc analysis, the vividness was significantly greater in the mother condition than in the father condition (t = 2.76, p = 0.011). The response intensity showed significant results in a main effect of parent (F1,23 = 24.23, p < 0.001) and a main effect of situation (F2,46 = 6.83, p = 0.003), but not in an interaction effect (F2,46 = 1.45, p = 0.246). Post-hoc analysis revealed that the response intensity was significantly higher in the mother condition than in the father condition (t = 4.92, p < 0.001) and significantly higher in the positive situation and negative condition than in the neutral situation (t = 3.24, p = 0.004; t = 3.95, p = 0.001, respectively). The arousal showed significant results in a main effect of parent (F1,23 = 6.66, p = 0.017) and a main effect of situation (F2,46 = 5.62, p = 0.007), but not in an interaction effect (F2,46 = 0.98, p = 0.384). In post-hoc analysis, the arousal was significantly higher in the mother condition than in the father condition (t = 2.58, p = 0.017), and in the positive and negative conditions than in the neutral condition (t = 3.35, p = 0.003; t = 3.25, p = 0.004, respectively).

Reviewer #2: 

Thank you very much for the possibility to review the manuscript titled "The neural influence of autobiographical memory related to the parent-child relationship on psychological health in adulthood". I think that this study contributes a lot to the literature and should be published in this journal if the authors want to review some aspects of the article that are lacking. This study investigates the neural influence on autobiographical memories; however, the introductory section refers to several concepts not explored in this work: emotional regulation, parent-child attachment. I think it may be important to review the whole introductory section in order to better focus the present work. In fact, the authors are invited to review studies related to the parent-child relationship (rather than to the model of indivisual attachment) and to deepen the literature on autobiographical memory.

For example, the authors are recommended to deepen their knowledge of the following works, which would greatly enrich this manuscript:

Fivush, R., & Haden, C. A. (Eds.). (2003). Autobiographical memory and the construction of a narrative self: Developmental and cultural perspectives. Psychology Press.

Fivush, R. et al. (1996). Remembering, recounting, and reminiscing: The development of autobiographical memory in social context.

Cimino, S. et al. (2020). Dialogues about Emotional Events between Mothers with Anxiety, Depression, Anorexia Nervosa, and No Diagnosis and Their Children. Parenting, 20(1), 69-82.

Reply:

We would like to express our gratitude for your optimistic evaluation of our research and for drawing our attention to an important point. In the revised manuscript, descriptions of emotion regulation and attachment were deleted because they were not directly related to the research purpose. Instead, we added additional information to the introduction to clarify autobiographical memory-related contents, including the references that you suggested. Consequently, the introduction was entirely rewritten.

Revision:

In ‘the 1st to 3rd paragraphs’ of the Introduction section

Like the saying, “no health without mental health,” promoted by the WHO, psychological health or psychological well-being is a fundamental factor contributing to quality of life [1]. There are many factors that influence psychological health in adulthood, and one of the most central determinants is the parent-child relationship [2]. The relationship between a parent and child becomes the template for other interpersonal relationships because attachment bonding allows a person to understand their environment and establish a sense of security [3,4]. An examination of the association between the retrospective perceptions of the parent-child relationship and emotional health in adulthood showed that daily emotional and psychological distress are closely related to how one perceives the quality of the parent-child relationship, and the retrospective perceptions of father-child and mother-child relationships contribute to different dimensions of emotional health [5]. Therefore, the perceptions of the childhood relationship with parents are important in the overall well-being, even as adults.

The retrospective perceptions of the parent-child relationship are based on an individual’s autobiographical memory. The development of autobiographical memory is a social process that is influenced by the reminiscing style of parents, and thus an integral part of the developing sense of self and interpersonal relatedness [6,7]. Autobiographical memory represents the self and shapes emotional states of our everyday lives [8,9]. In general, memories are highly intertwined with features like self-referential processing and emotion [9]. Functional neuroimaging studies have examined autobiographical memories in reference to other types of memories. For instance, a recent study that compared the difference between autobiographical memory and simple recognition memory found that when a task required retrieval of life memories, default-mode areas were engaged more, whereas a simple recollection of memories engaged the parietal memory network [10]. Since autobiographical memory retrieval relies heavily on self-reference and emotion, retrieval of autobiographical memories has been shown to employ various cortical and subcortical regions, including the prefrontal and parietal cortices, anterior cingulate cortex (ACC), precuneus, medial temporal cortex, hippocampus, and amygdala [11-14]. In particular, the amygdala has been implicated in the encoding of emotional memories [15] as well as retrieval of autobiographical memories [16]. Previous studies also found coactivation of the amygdala, hippocampus, and inferior frontal gyrus (IFG) during the retrieval of autobiographical memories [17] and neural responses in the dorsal to caudal ACC and IFG elicited by the retrieval of painful autobiographical memories [18]. These reports consistently show a close relationship between autobiographical memory and emotional memory.

Psychological health is very closely related to memory encoding and retrieval, and there is evidence supporting its influence on neural activities related to autobiographical memory. Numerous studies have also shown the connection between psychological health and selective memory bias. The representation, recall, and maintenance of autobiographical memories have shown to be disturbed by individuals with poor psychological well-being, where they are systematically biased for negative information [11]. Memory bias and negative interpretations of events contribute to a vicious cycle, where one’s psychological health contributes to the tendency toward negative interpretation of memories, and this bias also maintains low self-esteem, depression, or anxiety [19-21]. In fact, there is a consensus that self-esteem, anxiety, and depression are core elements of psychological health [22-24]. The importance of these elements has spurred researchers to study the close associations between self-esteem, anxiety and depression, and quality of life in numerous fields [25-28].

Again with regard to the introductory section, the authors are invited to define more clearly the objectives of the work and the hypotheses, consistent with the results and discussions.

Reply:

As suggested, we revised the objective and hypothesis.

Revision:

In ‘the last paragraph’ of the Introduction section

In this study of the general population, we sought to identify the neural substrates of the impression of parents while participants recalled childhood memories. For this purpose, we examined how fathers and mothers are perceived differently and whether the valence of memory influences this perception, and surveyed the effects of psychological health (self-esteem, depression, and anxiety) on the memory process. As childhood memories of parents triggered by various situations are inherently charged with emotions, we hypothesized that regions often involved in an autobiographical memory, such as the ACC, IFG, amygdala, and hippocampus, would be engaged differently depending on the parent and the valence of situation, and would be modulated by a person’s psychological health.

Discussions can also be reviewed in the light of the cited literature and authors are invited to use an editing service for English language.

Reply:

As suggested, we have completely revised the discussion and the manuscript was reviewed by a native speaker for English expression.

Revision:

In ‘the 1st to 8th paragraphs’ of the Discussion section

The present study examined the neural relationship between the perception of a parent and the valence of an autobiographical memory, then explored how it is modulated by one’s psychological health. We measured perception by inciting imagination of the facial expression a parent had made in a given situation. The three seconds given for the task may have been too short to remember an autobiographical memory. However, since the vividness and reaction intensity in the stimulus assessments were rated as moderately or higher, we can say that the task was to evoke an autobiographical memory rather than an imagination or a simple impression. From our behavioral data, we saw a rank in imagined emotion scores among the valences of childhood memories, where positive situations elicited the highest imagined emotion score, followed by neutral situations and negative situations. In conjunction with the RT results, we could infer a sense of hesitation in choosing an appropriate expression especially for negative childhood situations compared to positive situations. 

The quality of communication between parents and children is an excellent indicator of the quality of a parent-child relationship. Studies suggest that open communication with parents allows children to learn skills that are closely related to healthy coping skills and problem-solving strategies [54,55]. Thus, we assessed whether the quality of the relationship with parents affects the impression of parents during childhood. As expected, we found that more open communication with either father or mother was correlated with a greater emotion score in positive emotion situations for both parents. On the other hand, negative correlations were seen between communication problem scale scores (father/mother) and the imagined emotion ratings for the respective parent-positive conditions. Altogether, these findings convey that relationship quality directly affects the recollection of one’s parents in positive contexts, but not necessarily in negative situations. Moreover, the results also revealed that one’s level of self-esteem positively influences the perception of one’s father, particularly in positive situations. This finding further highlighted the influence of self-esteem on how positively one perceives his or her father for positive memories, rather than rumination on exaggerated negative affects seen in previous patient studies [19,41]. 

From the neuroimaging data, we saw intriguing neural responses to each condition from the right caudal ACC and precuneus, in particular. In the result of the caudal ACC, there was a particular bias toward father-negative and mother-positive, suggesting that in the autobiographical memory of emotional events, different forms of brain reactions occur depending on the parent's gender. Attention on the caudal ACC has risen in recent years and studies have indicated that the region contributes significantly to negative affect and pain processing [56-58]. This region receives an ample amount of emotion-related and voluntary movement signals [59,60], and has been suggested as a pivotal site for the emotional and voluntary interaction [58]. In a fascinating investigation on pain processing in self and others, it was seen that the caudal ACC was activated in response to self-experienced pain, but not when perceiving pain in others [56]. Furthermore, this region was shown to increase in response to negatively charged autobiographical memories [61]. On the other hand, its involvement in other-experienced pain processing has also been reported. A meta-analysis on empathy-related regions indicates that this region is part of the pain empathy network, and is engaged when pain was anticipated [62]. Regardless, it is clear that the caudal ACC plays a role in processing negative affect.

In light of these caudal ACC functions, our results may be linked to mothers taking on the role of primary caretaker more so than the fathers. In many cultures, traditional gender roles have determined the roles and expectations for parents, which usually assumes fathers as breadwinners and mothers as the primary caretaker in child rearing [63,64]. This point can be related to our results of the stimulus assessment. The analysis of the stimulus assessments indicated that the responses that fathers gave were less intense in all situations than those given by mothers; and the imagination of fathers’ appearances were less vivid and evoked less arousal than that of the mothers. In these respects, it is possible that caudal ACC activity may be a response to the stoic and emotionally distant perception of fathers. Meanwhile, the importance of memory to the mother can be seen in our result of negative correlation between caudal ACC activity in the mother-positive condition and depression score. There was a report that pathological changes in this region may be related to major depressive disorder [65]. A study on the behavioral adaptation to unpleasant environment showed the involvement of the caudal ACC to be critical in the ability to regulate the behavior [66]. It has been suggested that the region might contribute to defending one from developing depression [67]. Taken together, our result of the negative correlation in the mother-positive condition may suggest that through the function of the caudal ACC, autobiographical memories of positive events with the mother may act as a factor that prevents depression.

The precuneus, a central hub of the default mode network, also exhibited a similar interaction effect between the parent and situation during the childhood memory recollection task. The precuneus has been thought to play a critical role in episodic memory [68,69], source monitoring [70], and self-referential processing [71]. Recently, a multivariate analysis method, representational similarity analysis, demonstrated that the precuneus is a key region engaged during the subjective experience of vivid autobiographical memory reminiscence [72]. Other experiments have confirmed its prominent involvement in the vividness of self-related memories [73,74]. In addition, right lateralization of the precuneus has been shown for highly personal and vivid reminiscence of autobiographical events [72]. Right lateralization is corroborated by the present study, which required participants not only to visualize a given circumstance, but also to delineate the facial expression by either parent. Our results indicate overall greater activation of the right precuneus during recollection of fathers than mothers, and more specifically, its activity was higher when recollecting fathers in negative compared to positive situations. It is interesting to note that despite lower vividness score in the stimulus assessments for fathers than mothers, these findings were observed, suggesting that the perceptions of fathers may be more personal than those of mothers, especially during reminiscence of negative situations. These may indicate the saliency of childhood memories of fathers and particularly how fathers responded during negative circumstances. In conjunction with activity of the caudal ACC, we cautiously conclude that memories of fathers in negative situations have a more adverse effect as the negative impression of fathers is more permanently embedded. This interpretation can also be linked to the results of correlation with self-esteem.

 Studies report that the neural structure of personal semantic memories serves as a neural correlate of self-esteem [75,76]. It is postulated that due to its role in using autobiographical memories to choose the applicable self-relevant information, activation of the precuneus is closely related to self-esteem and how one processes given information [76,77]. The current study confirmed the established association between self-esteem and precuneus activity, as we found a negative relationship between one’s self-esteem and precuneus activity during autobiographical reminiscence of fathers in negative events. Recently, the precuneus was identified to correlate positively with evaluation sensitivity, measured by a change in state self-esteem [75]. Together with the evidence supporting the precuneus’ affiliation with evaluation sensitivity, as well as the tendency for individuals with lower self-esteem to be more affected by negative information [76], our result seems to suggest that the heightened activation observed in individuals with lower self-esteem may be due to the saliency of negative memories during recollection and their tendency to be more influenced by negative information. 

 Our PPI analysis revealed significant increases in functional connectivity of the caudal ACC with the fusiform gyrus and postcentral gyrus for the father-positive condition compared to the mother-positive condition. Given that the fusiform gyrus processes facial identity [78] and emotion [79], our finding may indicate that a greater sense of negative affect was involved while processing facial expressions of fathers than mothers in positive childhood events. This raises the possibility that the negative affect arisen from processing emotional face expressions may be particularly different between the parent in a positive context. Going back to the stimulus assessments, we saw lower vividness, response intensity, and arousal from fathers than mothers. Again, this may be since, generally, mothers are primary caregivers who spend more time with children; thus, one may have more readily available positive memories involving mothers than fathers to access on demand.

Previously, individuals with high trait anxiety have shown to hyperactivate the fusiform gyrus while processing uncertain cues and angry faces [80,81]. Consistent with the activation study, our study revealed that the connectivity strengths between the cingulate cortex and fusiform gyrus during the father-positive versus mother-positive condition correlated positively with the severity of anxiety symptoms. These results may provide the neurobiological influence of trait anxiety in the retrieval of memory and the perception of fathers. In addition to emotional face processing, greater connectivity between the cingulate area and postcentral gyrus was witnessed for the father-positive than mother-positive condition. Functional connectivity between these two regions has been implicated in posttraumatic stress disorder (PTSD), where patients with PTSD showed stronger connectivity than the control group, and also was positively correlated with the severity of symptoms [82]. As part of the sensorimotor network [83], the coupling of the caudal ACC and postcentral or paracentral gyri has been shown to be involved in shock processing [84] and other emotional memory tasks [85]. Based on the evidence, we postulate that the caudal ACC may relay negatively valenced affective information to the sensorimotor area more for fathers than mothers when processing facial expressions.

---

## [Editor Report · Decision Letter 1]

27 Mar 2020

The neural influence of autobiographical memory related to the parent-child relationship on psychological health in adulthood

PONE-D-19-30869R1

Dear Authors,

We are pleased to inform you that your manuscript has been judged scientifically suitable for publication and will be formally accepted for publication once it complies with all outstanding technical requirements.

With kind regards,

Luca Cerniglia, PhD

Academic Editor

PLOS ONE

Additional Editor Comments (optional):

The authors were responsive to all comments. I think the manuscript can be published in the present version.
---

## [Editor Report · Acceptance letter]

31 Mar 2020

PONE-D-19-30869R1 

The neural influence of autobiographical memory related to the parent-child relationship on psychological health in adulthood 

Dear Dr. Kim:

I am pleased to inform you that your manuscript has been deemed suitable for publication in PLOS ONE. Congratulations! Your manuscript is now with our production department. 

With kind regards,

on behalf of

Dr. Luca Cerniglia 

Academic Editor

PLOS ONE